# Detangling electrolyte chemical dynamics in lithium sulfur batteries by operando monitoring with optical resonance combs

Fu Liu[1,2], Wenqing Lu[3], Jiaqiang Huang [4], Vanessa Pimenta[3], Steven Boles [5], Rezan Demir-Cakan [6,7] ✉ & Jean-Marie Tarascon [1,2,8] ✉

Challenges in enabling next-generation rechargeable batteries with lower cost, higher energy density, and longer cycling life stem not only from combining appropriate materials, but from optimally using cell components. One-size-fits-all approaches to operational cycling and monitoring are limited in improving sustainability if they cannot utilize and capture essential chemical dynamics and states of electrodes and electrolytes. Herein we describe and show how the use of tilted fiber Bragg grating (TFBG) sensors to track, via the monitoring of both temperature and refractive index metrics, electrolyte-electrode coupled changes that fundamentally control lithium sulfur batteries. Through quantitative sensing of the sulfur concentration in the electrolyte, we demonstrate that the nucleation pathway and crystallization of Li$_2$S and sulfur govern the cycling performance. With this technique, a critical milestone is achieved, not only towards developing chemistry-wise cells (in terms of smart battery sensing leading to improved safety and health diagnostics), but further towards demonstrating that the coupling of sensing and cycling can revitalize known cell chemistries and break open new directions for their development.

Wide-scale utilization of renewable energy sources is essential to supplementing and perhaps replacing the carbon-based energy supply responsible for climate change. The recent success of electric vehicles made possible by lithium-ion battery technology, attributed to both improved reliability and cost reductions, demonstrates that new breakthrough chemistries may not be necessary if known electrochemical cell pairings can be mastered. Included among these chemistries are resurgent lithium sulfur batteries (LSB), which, in spite of their appeal in terms of theoretical specific energy (~2600 Wh/kg), are still not commercialized. This can be attributed to a number of unresolved challenges, including the insulating nature of sulfur and lithium sulfides, large volume expansion (80%) of the solid sulfur cathode during the formation of Li$_2$S, and the shuttle effect caused by soluble polysulfide in electrolyte[1,2].

Numerous characterization techniques have been deployed to clarify the underlying science of LSBs during operation, which have contributed significantly to a better understanding of the kinetics and thermodynamics of the dissolution/precipitation of polysulfides, whose critical role in LSBs has been known for nearly 50 years[3]. Since then, methods such as X-ray diffraction (XRD)[4,5], electrochemical tests[6–8], and spectroscopic techniques[9–16] have been used to provide valuable information regarding identification of polysulfide species

[1]Collège de France, Chimie du Solide et de l'Energie—UMR 8260 CNRS, Paris, France. [2]Réseau sur le Stockage Electrochimique de l'Energie (RS2E)—FR CNRS 3459, Amiens, France. [3]Institut des Matériaux Poreux de Paris (IMAP), ESPCI Paris, Ecole Normale Supérieure, CNRS, PSL University, Paris, France. [4]The Hong Kong University of Science and Technology (Guangzhou), Sustainable Energy and Environment Thrust, Nansha, Guangzhou, Guangdong 511400, P. R. China. [5]Department of Energy and Process Engineering, Faculty of Engineering, Norwegian University of Science and Technology (NTNU), Trondheim, Norway. [6]Institute of Nanotechnology, Gebze Technical University, Kocaeli 41400, Turkey. [7]Department of Chemical Engineering, Gebze Technical University, Kocaeli 41400, Turkey. [8]Sorbonne Université–Université Pierre-et-Marie-Curie Paris (UPMC), Paris, France. ✉e-mail: demir-cakan@gtu.edu.tr; jean-marie.tarascon@college-de-france.fr

and reaction kinetics. However, it is experimentally challenging to isolate the individual polysulfides due to the propensity of disproportionation, and these analytical techniques rely on special equipment and cell designs that cannot be directly deployed for long cycling periods. Recently, optical fiber sensors have attracted attention in battery sensing due to their low cost, compactness, remote sensing capabilities, and simple integration into batteries without interfering with their internal chemistry[17]. Among the fiber sensor family, the most commercialized Fiber Bragg grating (FBG) sensors have been well integrated inside Na (Li)-ion batteries for monitoring heat and pressure[18] or inside the solid-state batteries for tracking the stress dynamics[19]. Indeed, recently Ziyun et al. demonstrated that the cathode stress evolution of LSB can be in-situ monitored by FBG sensors for understanding the chemo-mechanics[20]. Nevertheless, tracking polysulfides with FBGs is still limited, owing to the fact that the sensing signals are totally confined inside the fiber core and cannot sense the electrolyte surrounding the fiber surface.

In order to investigate the external medium of fiber, tilted fiber Bragg gratings (TFBGs, same structure as FBG without physical structure modification[21], but rotating the grating plane to a specific angle) have been proposed to excite hundreds of discrete cladding mode resonances that are sensitive to the external medium refractive index perturbation via evanescent fields[22], hence serving as an optical comb. This has led to the development of high-performance sensors used in various areas, including biomedicine[23], magnetic detection[24], and gas monitoring[25]. Recently, TFBGs have been integrated into commercial batteries to detect chemical dynamics/state of electrolytes related to chemical evolution[26]. Interestingly, some TFBG-assisted surface plasmon resonance (TFBG-SPR) sensors with higher sensing sensitivity have also been developed for Zn-ion batteries to offer an alternative way of probing ion transport kinetics[27]. Overall, TFBG sensors provide new opportunities to deal with the challenge of battery sensing as they combine direct optical chemical sensing, as well as physio-mechanical parameters via the confined optical modes.

Herein, TFBG sensors, enabling measurements with a wide array of parameters including refractive index, temperature and strain, are proposed to operando track the chemical dynamics/states of the LSB via electrolyte sulfur concentration. We demonstrate that the capacity fading is strongly correlated with the dissolution/precipitation of polysulfides throughout cycling and hence, with respect to cycling rates. By exploiting the kinetic and thermodynamic responses of soluble sulfur in the electrolyte, the nonlinear net transport flux clarifies the invisible disproportionation process and the origins of its dynamic evolution. With this understanding, we show that altering the nucleation pathway of the crystalline $Li_2S$ and sulfur can be attributed to real improvements in cell cycling performance. Subsequently, it is noted that TFBGs have the ability to obtain key chemical-physical-thermal metrics *in operando* with notable time and spatial resolution that may extend beyond LSBs.

## Results

### Characteristics of TFBG sensing

Prior to *in operando* battery inspection, it is appropriate to first briefly visit the suitability of TFBG sensing for such chemistries, as related to fundamental principles of their operation. TFBGs, immersed in an electrolyte (Fig. 1a), were made in the core of the commercial single-mode fiber by ultraviolet pulse laser to induce periodically permanent refractive modulation. They obey a phase matching condition by enhancing the coupling between fundamental core mode and backward-propagation cladding modes[22] (Fig. 1b):

$$\lambda = (n_{11}(\lambda) + n_{lm}(\lambda))\Lambda / \cos\theta \qquad (1)$$

where $\lambda$ is the cladding mode resonance wavelength, $n_{11}(\lambda)$ is the effective index of core mode, and $n_{lm}(\lambda)$ is the effective index of

cladding mode with azimuthal order $l$ and radial order $m$. $\Lambda$ is the period of grating along the fiber axis, and $\theta$ is the grating tilt angle. The experimental spectra are presented in Fig. 1c, where the core mode resonance (i.e., Bragg resonance) is located at the longest wavelength around 1590 nm (sensitive to temperature and strain ($T$, $\varepsilon$))[22]. The cladding mode resonances guided by the fiber cladding (beside $T$, $\varepsilon$, also sensitive to refractive index ($RI$) of the surrounding media) are shown on the left of Bragg resonances. The leaky modes are located at the region where there is a discontinuity in the cladding mode envelope, indicating the loss of total internal reflection at the point where the cladding mode effective index becomes equal to or smaller than the surrounding $RI$. Therefore, with respect to soluble polysulfides which perturb electrolyte density, and hence the refractive-index, we focus on the cutoff guided cladding mode near the leaky mode region (around 1560 nm wavelength) which is insensitive to unpolarized input light (i.e. can be probed without a polarizer, which simplifies sensing system and still ensures that detection is both stable and repeatable) and shows the highest refractive index sensitivity[22].

To investigate the response of TFBG to polysulfides, depicted in Fig. 1c, it was thoroughly immersed in a series of 100 mM polysulfide containing electrolytes in a modified Swagelok cell. Bearing this in mind, the Bragg resonance remains stable because any strain and temperature variation were eliminated during the measurements, indicating that the cladding mode wavelength shift is only related to refractive index variation. When the chain length of polysulfides is increased while keeping the polysulfide concentration the same, the guided modes on the left side of cladding mode at 1560 nm become leaky due to the increased refractive index. This is a result of the number sulfur atoms in solution becoming larger and perturbing the corresponding mode effective refractive index, while guided modes on its right side are linearly shifted to longer wavelength (Fig. 1d, e and Supplementary Fig. S2a, b). Noteworthy is the fact that the refractive index tested by TFBG sensor is an average effect of all the pertinent refractive indices of lithium polysulfide solutions. Following this, dilution of the 100 mM polysulfide $Li_2S_x$ ($x = 4, 5, 6, 7, 8$) to an equivalent concentration of sulfur (Supplementary Fig. S2c) yields an equivalent optical effect, stemming from the refractive index of polysulfide solutions converging to the same density, (Fig. 1f and Supplementary Fig. S2d, e). Therefore, rather than recognizing the specific species inside the electrolyte, the TFBG sensor will distinctly reveal the electrolyte sulfur concentration evolution of LSB cell.

### Operando measurement of chemical dynamic state of LSB

Given the promising proof-of-concept of sulfur concentration measurement in electrolyte, we explored the capability of operando chemical dynamics/states testing by putting a 2 mm thick, 12.8 mm diameter polyether ether ketone (PEEK) ring (such that a 1 cm long TFBG can pass through) in the middle of the Swagelok assembly to separate the cathode (sulfur and Super P carbon composite (60/40 wt.%)) and anode (lithium). This configuration ensures that the fiber sensor would not touch either electrode (Supplementary Fig. S3), and thereby avoiding the strain induced by cathode, which are known to exhibit around 80% volume changes during cycling[20]. Meanwhile, any thermal effect stemming from the electrolyte background environment has been calibrated and compensated by the method detailed in Supplementary Fig. S4. Filling the PEEK ring with electrolyte (250 μL, 1 M LiTFSI, 0.5 M LiNO$_3$ in DOL/DME (1:1, v/v)) where the sensor is immersed, the effect of ion concertation gradient of electrolyte including Li$^+$, TFSI$^-$ and NO$_3^-$ in DOL and DME can be safely neglected. It should be noted that prior to LSB investigations, a control experiment was executed with lithium iron phosphate (LFP) as the cathode. Here it was found that the corresponding $RI$ of electrolyte variation is 20 times smaller than that in LSB in which dissolved polysulfide is formed (Supplementary Fig. S5). Therefore, the use of a TFBG can provide the possibility of measuring sulfur concentration in the electrolyte during

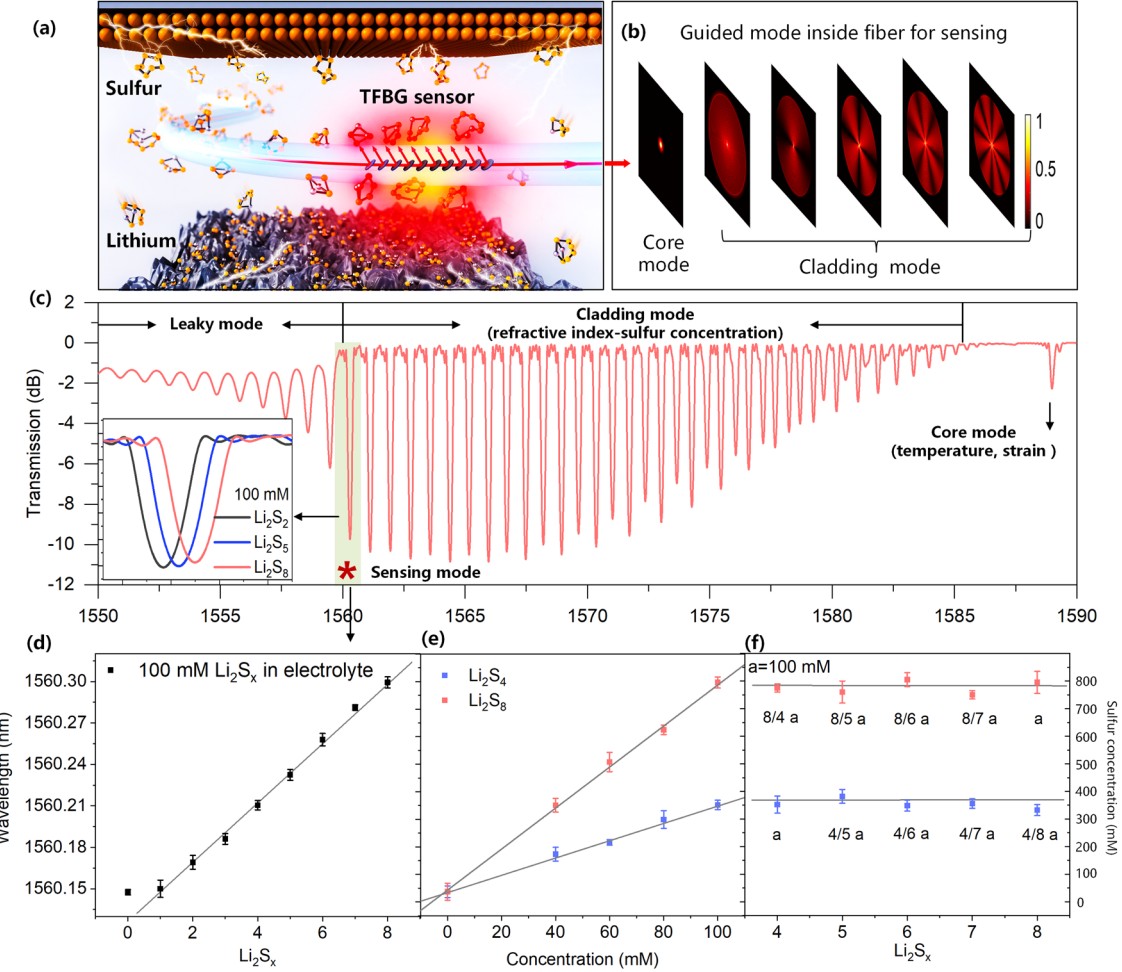

**Fig. 1 | Concept of optical fiber sensing for LSB. a** Schematic of a fiber optic sensor immersed in electrolyte for in-situ detection of sulfur concentration originating from the generated dissolved polysulfide and their transport activities (i.e., shuttle effect). **b** Backward-propagation guided modes inside fiber for sensing (Supplementary information Fig. S1). **c** Experimental spectra response to polysulfide. **d** The wavelength shifts of cladding mode resonance at ~1560 nm to 100 mM polysulfide $Li_2S_x$ ($x = 1, 2, 3, ..., 8$), shaded in green; (**e**) to concentration variation of $Li_2S_4$ and $Li_2S_8$ from 0 mM to 100 mM; (**f**) to same sulfur concentration of polysulfide $Li_2S_x$ ($x = 4, 5, 6, 7, 8$). The error bars represent the measurement error (test 3 times continuously) resulting from the surrounding temperature change and electrolyte solvent evaporation.

cell operation, and to a large extent, the measurement will be irrespective of the cell's state of charge or state of health.

Based on the aforementioned concept, we measured the electrolyte sulfur concentration variation with a TFBG sensor while simultaneously deploying *in operando* XRD to track the phase transition of the composite electrode (Fig. 2a). At the upper voltage plateau around 2.4 V, the highest sulfur concentration in the electrolyte was monitored (left panel of Fig. 2a) and found to be accompanied by a decrease in the sulfur peak intensity (XRD pattern) resulting from a series of phase transformations, i.e., from solid sulfur to soluble intermediate polysulfides. On the other hand, when the carbonate-based electrolyte was used as a reference (i.e., LP30, right panel of Fig. 2a), no concentration gradient was observed in the electrolyte and remained nearly stable due to the fact that no soluble polysulfide intermediates were formed. Instead, this resulted in the formation of insoluble and undetected products, since it is known that there is a nucleophilic reaction between sulfur radical and ethylene carbonate of LP30 to form thiocarbonate-like solid electrolyte interphase[28,29] (Supplementary Fig. S6). Turning to the lower voltage plateau around 2.1 V (Fig. 2a), the concentration of dissolved sulfur decreases as a result of the reduction of long-chain polysulfide into shorter chains, leading to insoluble $Li_2S$ compound in the cathode (Supplementary Fig. S7a) and verified by XRD[30,31]. Upon charging, the sulfur concentration indicates

reversible recovery consistent with the decay of $Li_2S$ peaks until complete disappearance at the voltage ~2.4 V, where crystallization of sulfur starts and thus sulfur concentration in electrolyte drops again, even though the deposited solid sulfur in the cathode is featureless by XRD[31]. To identify the sulfur at the end of charge, the cathode powder was recovered in the glovebox by washing and drying to remove any soluble polysulfide as well as remaining electrolyte salts (Fig. 2b, c), confirming that recrystallized sulfur was detected and its surface topography was unchanged (i.e., presence of amorphous sulfur)[31]. Furthermore, when setting the 15-h open circuit voltage (OCV) after charging, the sulfur concentration increases and reaches a plateau within 9 h (Fig. 2c), whereas, on the other hand, no sulfur concentration changes were observed during rest periods applied at the end of discharge (Supplementary Fig. S7b). It is most likely explained by comproportionation reactions during the rest period when the recrystallized sulfur from the end of charging is transformed to soluble lower-order polysulfide via reacting with high-order polysulfide[32]. This is also supported by the beginning of 2 h rest (first cycle before discharging) demonstrating relatively little variation of electrolyte since fresh electrolyte contains a minimum amount of high order polysulfide and the comproportionation reactions are thus not possible (Supplementary Fig. S7c). The crystalline sulfur at the end of charge is related to the cut-off potential that the sulfur recrystallization process

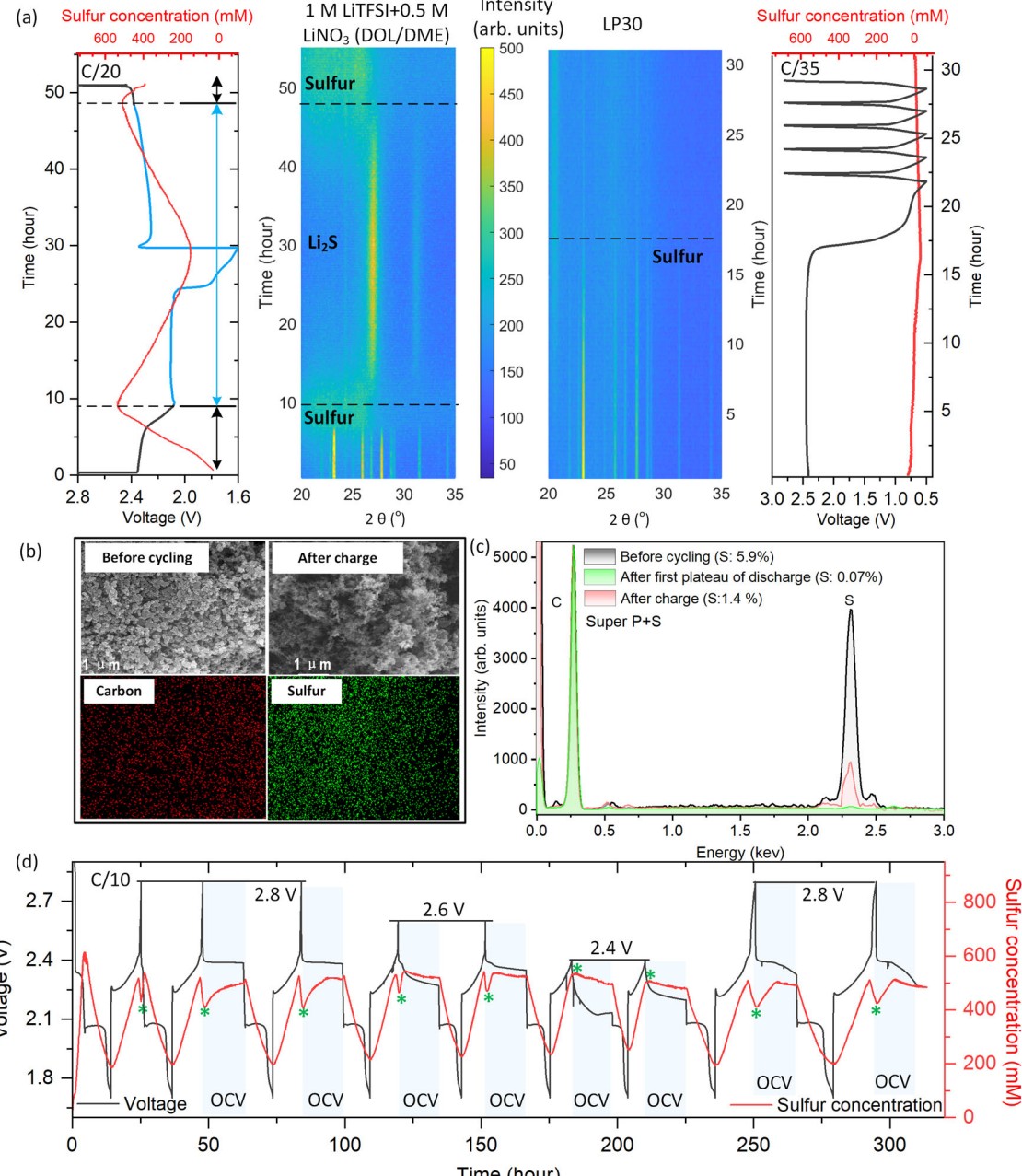

**Fig. 2 | Decoding electrolyte sulfur concentration dynamic of LSB. a** by TFBG and XRD at C/20 (0.275 mA left panel): polysulfide dissolution allowed (electrolyte of 1 M LiTFSI, 0.5 M LiNO$_3$ in DOL/DME (1:1, v/v)); right panel: polysulfide dissolution prohibited (electrolyte of LP30: 1 M LiPF$_6$ in EC/DMC) of sulfur and Super P carbon composite (60/40 wt.%) electrode. **b** Morphology (SEM) of cathode before cycling and after charge, and content of elemental sulfur and carbon (energy-dispersive X-ray spectroscopy, EDX) of the cathode after charge. **c** The quantitative analysis of sulfur before cycling, end of first plateau of discharge and end of charge. **d** The recrystallized sulfur governed by comproportionation reactions and cutoff voltage (C/10, 0. 533 mA). The shaded region in blue stands for 15 h of rest (OCV mode) starting at the end of charge demonstrating that the re-crystallized sulfur (marked by green asterisk "*") dissolves into the electrolyte in the form of soluble polysulfide through comproportionation reactions. All the data has been duplicated at least two or three times prior to being reported herein (Supplementary Fig. S7). The C-rate is defined by the speed at which a battery is fully charged or discharged. In order to better understand the C-rate, the absolute current with every instance of C-rate used was provided.

disappears[4] (disappearance of sulfur concentration valley at the end of charge in Fig. 2d) if setting the potential below 2.4 V (indicated that less sulfur suppresses the related comproportionation reactions, also detailed in Supplementary Fig. S7c).

Altogether, the dynamic of sulfur concentration of electrolyte decoded by TFBG sensor supports the simplified chemical reaction process: during discharge the sulfur receives electrons and transfers them first to soluble Li$_2$S$_4$ at the high voltage plateau. This is followed by formation of insoluble Li$_2$S at the low voltage plateau and vice versa

for the charging process, indicating that the consumption rate of sulfur under such galvanostatic conditions can be expressed as a ratio. According to the linear sulfur concentration variation rate, calculated from monitoring the slope (mM/h) on each plateau tested by the sensor during the discharge and charge steps (Supplementary Fig. S5a), it was observed that the ratio on the upper and lower plateaus of the discharge step is 3.88, while value obtained during charging is 0.84. It suggests that the rate of sulfur transformation to/from soluble polysulfide is 3.88 times faster than that to/from

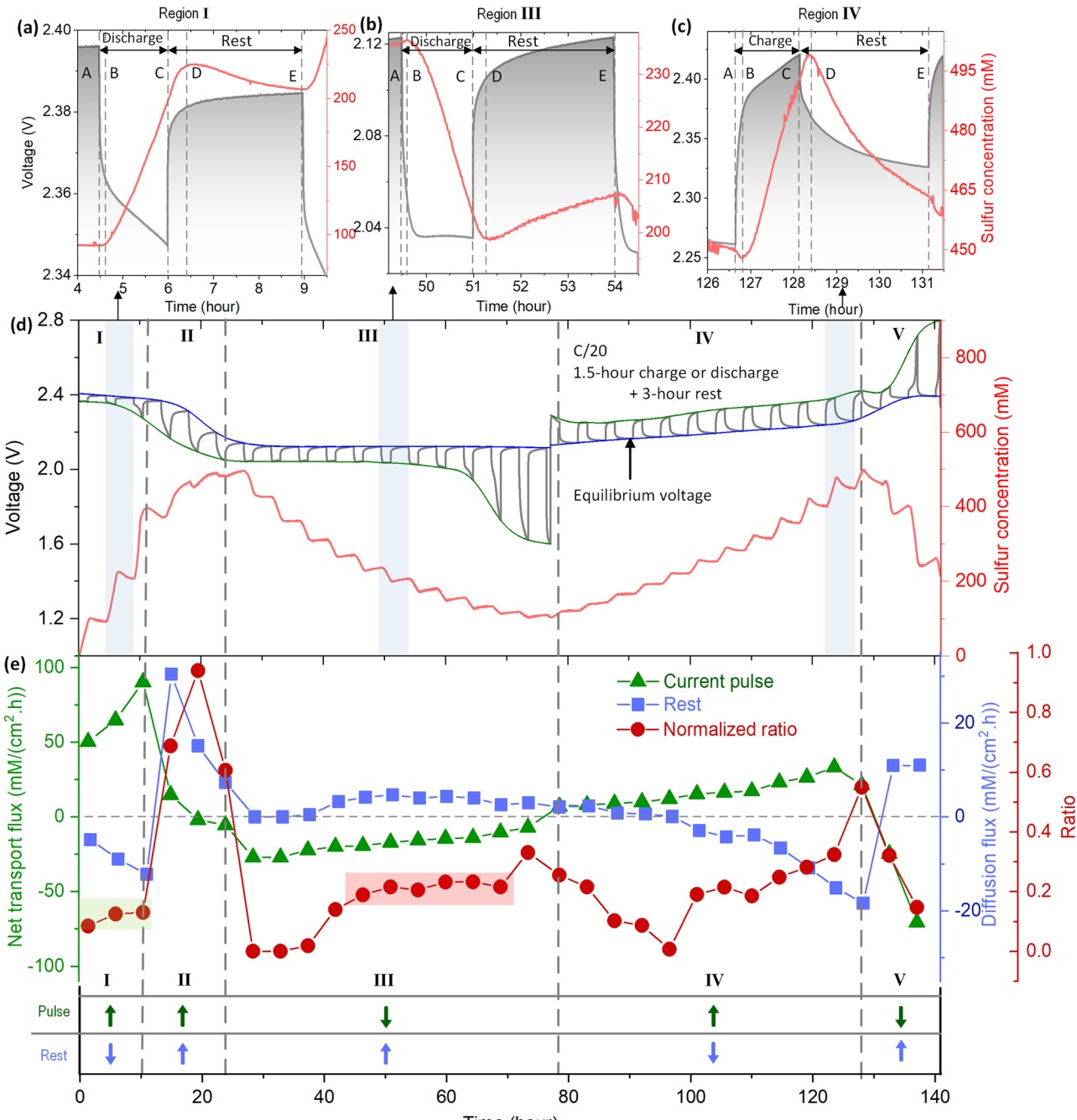

**Fig. 3 | Decoding the disproportionation process and evolution. a–d** The temporal voltage (black curve) and decoded electrolyte sulfur concentration (red curve) by GITT test (the capacity and optical spectra are given in supplementary Fig. S8). **a–c** Detailed view of electrolyte response to current pulse (kinetic process) and rest (thermodynamic process). **e** Net transport flux of soluble sulfur based on current pulse (kinetics process, green triangle, $D_k = V_{BC}/(S \times t)$) and rest (thermodynamic process, blue square, $D_t = V_{DE}/(S \times t)$), where $S$ is across section area of electrolyte that sensor is immersed in, $t$ is corresponding time. It indicates the slope of soluble sulfur consumption (negative) or formation (positive) in electrolyte, which is also plotted by arrows in bottom (arrow up: sulfur increasing, arrow down: sulfur decreasing). The normalized ratio (red sphere) represents the sulfur consumption of rest (thermodynamic process) by $Ratio = |D_t|/(|D_t| + |D_k|)$.

insoluble $Li_2S$ and the respective polysulfides for discharge (0.84 times for charge).

The disproportionation and association reactions of likely intermediates are mesmerizing, but despite an awareness of existence their mysterious dynamics make LSBs seemingly incomprehensible. Nevertheless, the real-time quantification of soluble sulfur afforded by TFBGs provides a convincing way to straighten their story when combined with GITT (Fig. 3). As depicted in Fig. 3d, the overall profile of sulfur concentration variation matches well with dissolution/precipitation of polysulfides and sulfur already confirmed in Fig. 2, and we

focus on the temporal response of the electrolyte to the current pulse and respective rest period. According to Fig. 3a–c, the tiny variation of sulfur concentration from A to B (moving from the rest to cycling mode) is the electrolyte instantaneous response to the leading edge of the current pulse (i.e. strong electric field gradient), originating from polysulfide redistribution driven by the sudden electrical field[33]. Meanwhile, Fig. 3a, b that concentration variation from A to B is opposite to that in Fig. 3c, this should be attributed to the change of the concentration gradient direction during discharge. We also note that the amplitude of concentration variation from A to B is much

smaller than that in Fig. 3c, resulting from the fact that the fiber sensor is physically/geometrically closer to sulfur electrode during assembly process, and therefore it leads to an asymmetric concentration variation. Overall, this effect trends in the opposite direction as polysulfide diffusion in the region from B to C, which relates to the current pulse (referred to herein as the kinetic process). The reader may note a discrepancy between voltage and concentration from C to D during rest, which is attributed to the delay between the electrochemical reaction at the electrode surface and the position of the sensor in the cell. Hence a small lag exists even if removing the current pulse. Regarding the OCV relaxation (3 h rest period) and movement towards equilibrium (herein coined as the thermodynamic process) in the region from C to E, the sulfur concentration fluctuation is most likely explained by polysulfide disproportionation process (i.e. $Li_2S_8 \leftrightarrow Li_2S_6 + 1/4S_8$)[9] since the two best-remaining hypotheses, dissociation (i.e. $Li_2S_6 \leftrightarrow Li^+ + LiS_6^-$ or $Li_2S_6 \leftrightarrow 2LiS_3$)[34] and non-uniform polysulfide distribution can be excluded: the dissociation of polysulfide including anions and radicals are rare while the neutral lithium polysulfide is dominant in the electrolyte[34]; the polysulfide distribution reaches equilibrium within 1 min (time interval of spectra recording) that is nearly synchronous to electrochemistry (Fig. 2a), not matching the rest period situation with 3 h of continuous soluble sulfur consumption or generation. After careful deliberation, we move forward with the idea that electrochemical and disproportionation processes can be extracted respectively by temporal response of electrolyte based on sulfur concentration decoded by TFBG sensor.

Encouraged by the results mentioned above, we have next attempted to build a quantitative relation between kinetic and thermodynamic processes through a primitive estimation of net transport flux of sulfur in the electrolyte (Fig. 3e). In STAGE I, on the higher voltage plateau, solid sulfur was continuously consumed upon discharge to form long chain polysulfide, $Li_2S_8$ (green triangle), together with the rapid disproportionation process $Li_2S_8 \leftrightarrow Li_2S_6 + 1/4S_8$[9,35–37] leading to soluble sulfur consumption during relaxation (blue square). At the same time the normalized ratio (red sphere) between the rest and current pulse process nearly remains the same and fixed at 0.1 (shaded in light green color), meaning that there is a competition reaction between the soluble long chain polysulfide species formation and the $S_8$ reprecipitation in the beginning of the discharge step. In STAGE II, regarding the first-to-second plateau transition whereby a voltage slope forms between 2.3 and 2.1 V, the shorter chain polysulfide $Li_2S_4$ is expected to be generated[38,39]. This is accompanied by a reduction in the rate of formation of soluble sulfur in the electrolyte and hence, the sulfur concentration during rest keeps increasing while the generation of fresh long chain polysulfides winds down and the kinetic/thermodynamic ratio can reach 0.89. This indicates that polysulfide species formation via disproportionation is dominant due to fact that dissolved sulfur continues to react with polysulfide presenting in the electrolyte during rest[38,40]. Undoubtedly an enriched concentration of sulfur in the electrolyte contributes significantly towards driving the formation of more polysulfides. In STAGE III where the lower voltage plateau marks the conversion between $Li_2S_4$ to shorter chain $Li_2S_2$ and $Li_2S$ forms, the potential disproportionation process $Li_2S_2 \leftrightarrow 1/3Li_2S_4 + 2/3Li_2S$[40,41] is highlighted from the middle of the second plateau, and raises the sulfur concentration to about a normalized ratio of 0.2 which follows until the end of the half-cycle. Upon charging (STAGE IV and STAGE V), it is evident that the process is not a fully reversible one, as seen with STAGE IV where $Li_2S_4$ and $Li_2S_6$ reappear and the push towards thermodynamic equilibrium necessitates disproportionation processes leading to consumption of sulfur during rest periods, but an overall concentration increase. In STAGE V, the sulfur concentration in electrolyte drops very sharply, caused by the recrystallization of sulfur during current pulsing, even though that is not visible in the operando XRD studies shown in Fig. 2a. Interestingly, the rise of soluble sulfur during these late-stage rest periods

suggests nucleation and/or growth limitations of the recrystallized sulfur, which will be addressed here later. Altogether, the quantitative disproportionation process decoupled by the fiber sensor based on GITT provides meaningful details to understand micro-mechanisms of complicated kinetics processes.

Inspired by aforementioned exploration of internal mechanisms of LSBs, we decide to further investigate the operando monitoring over cycling and cycling rates (Fig. 4a, b). Bearing in mind that the temperature (blue curve in Fig. 4b), decoded by Bragg resonance located at 1589 nm[26], initially rises to 25 °C and keeps a constant afterward due to the shipping from glovebox to oven to reach thermal equilibrium. The periodic sulfur concentration evolution (red curve in Fig. 4b), decoded by the wavelength shift of cladding mode located at ~1559.5 nm in Fig. 4a, indicates the reversible dissolution/precipitation of polysulfides and sulfur. Noteworthy here is the feasibility of observing the amplitude of soluble sulfur variation (supplementary Fig. S9) that matches the cycling behavior associated with capacity fading owing to less and less $Li_2S$ and solid sulfur crystallization over cycles (Fig. 4c), which could be reasonably attributed to the high electrolyte to sulfur ratio (E/S ratio > 100 μL/mg), thereby inducing stronger polysulfide shuttle effect with less sulfur utilization. Regarding a sulfur concentration response to the cycling rate depicted in Fig. 4d, a lower cycling rate (C/15) leads to the largest sulfur concentration change, strongly supporting the idea that the most soluble sulfur in electrolyte is transformed to solid species ($Li_2S$) when given adequate time for completion of the redox process. Accelerating the cycling to C/10, C/5 and C/3, partially soluble sulfur transformation is seen (i.e., background of sulfur concentration rises) together with less sulfur crystallization (i.e., the dip of soluble concentration at the end of charge).

Following the evidence thus far, it is then reasonable to conclude that regular galvanostatic operating conditions might not be perfect for LSBs, considering the various competing mechanisms explored above which are evident at different stages of charge and discharge. With this in mind, different modes of cycling protocols were also pursued to investigate the possibility of tuning cycling performance based on sulfur consumption and capacity. For instance, when setting a relatively low (high) cycling rate for the upper (lower) plateau depicted in Fig. 4e (Mode I), only half of the soluble sulfur inside electrolyte was transformed to $Li_2S$ due to high cycling rate of lower plateau, together with high rate of capacity variation. On the upper voltage plateau, the solid sulfur dissolution or recrystallization was 100% successful and which can be readily attributed to the low cycling rate in this voltage range. Further confirmation of the delicate cycling nature of LSBs was seen in Mode II as 80% of the soluble sulfur inside electrolyte was transformed to $Li_2S$ due to long cycling in the lower plateau (C/20) but with less solid sulfur recrystallization at the end of charge (C/5). Overall, the quantitative relation of soluble sulfur evolution and cycling (e.g. current density) implies that the efficiency of solid sulfur and $Li_2S$ formation governs the cycling performance and significant new progress for LSBs might be made through cycling condition optimization alone.

## High performance based on functional positive electrode

Extensive work has been conducted over the years to improve the LSB performance through variety of means: the use of high-conductivity sulfur electrodes, strong polysulfide binding to suppress the shuttling phenomenon, surface chemistry to control $Li_2S$ nucleation or dissolution, design of electrode architectures that are elastic to withstand volume expansion, and optimized electrolytes enabling high sulfur utilization[42]. Considering the strategy of trapping lithium polysulfides, it includes physical (spatial) entrapment by confining polysulfide in the pores of non-polar carbon materials[43] or designing a sulfur host material that exhibits stronger chemical interaction such as dipolar configurations based on polar surfaces[44], metal-sulfur bonding[45] and surface chemistry for polysulfide grafting and catenation[46], leading to

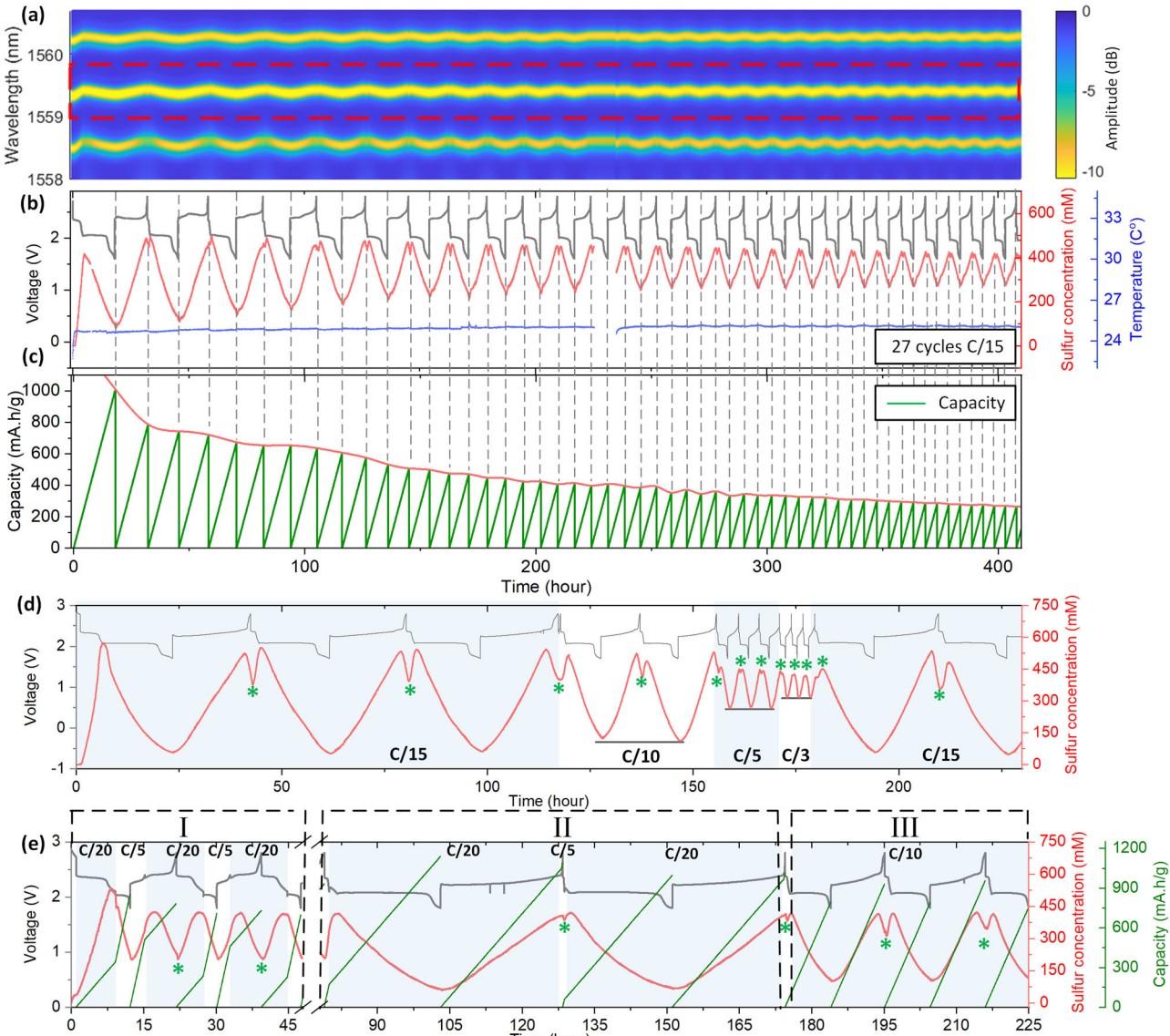

**Fig. 4 | Operando monitoring over cycling and cycling rate. a** Spectra contour of TFBG cladding mode resonances response (C/15, 0.361 mA, details are given in Supplementary Movie 1). **b** Temporal voltage (gray line), decoded sulfur concentration (red line), and temperature of electrolyte (blue line) over the cycling. Note that there is 10 h of data missing in 12th cycle because of data recording failure of the integrator software. **c** Capacity variation of (**b**) upon time. **d** Soluble sulfur concentration dynamic related to cycling rate (C/15, 0.368 mA; C/10, 0.553 mA; C/5, 1.106 mA; C/3, 1.502 mA). **e** Soluble sulfur concentration and capacity variations related to cycling rate of first/second plateau. The concentration drop through the re-crystallization of sulfur at the end of charge is marked by green asterisk "*". With Mode I, the cycling rate was set by upper plateau (C/20, 0.271 mA) and lower plateau (C/5, 1.081 mA); Mode II by upper plateau (C/5, 1.081 mA) and lower plateau (C/20, 0. 271 mA) and Mode III by upper plateau (C/10, 0.541 mA) and lower plateau (C/10, 0. 541 mA).

long cycling without strong capacity fading. Nevertheless, the fundamental problem (regardless of approach) is that polysulfides dissolution into the electrolyte is essential for LSB cells to function, and yet, the same polysulfide intermediates are responsible for deleterious parasitic reactions. Thus, a smart sensor that can reliably track the varying concentration of soluble sulfur in real time should be of particular value.

In this respect, in addition to the less porous Super P carbon discussed above, Ketjen black (KB) carbon, with a BET surface area >1200 m²/g, is utilized as physical nonpolar sulfur confinement host (KB/S, 40/60 wt.%), and the electrolyte was simultaneously monitored by a TFBG senor together with XRD to characterize the composite electrode phase transitions *in operando* (left panel of Fig. 5a). After the melt-diffusion treatment process, part of sulfur penetrates into the nanostructure of KB (Supplementary Fig. S10); however, during cycling the nonpolar physical adsorption or confinement of

polysulfide is very limited and massive soluble polysulfide is dissolved and detected. Surprisingly, the capability of Li₂S crystallization is enhanced since >91% soluble sulfur disappears and converts to solid short chain polysulfide at second plateau as indicated by the fiber sensor. Meanwhile, unlike linearly sulfur concentration change with super P carbon (BET surface area: 62 m²/g) substrate cell in Fig. 2a, KB containing cell becomes nonlinear with a higher rate, indicating that the nucleation rate of Li₂S is increasing. It could be reasonably assigned by instantaneous nucleation that depletion of the nucleation site of KB occurs at a very early stage, following the nonlinear kinetics of the nucleation pathway: $N = N_o[1-\exp(-At)]$ where $N$ is the density of nuclei, $N_o$ is the density of available nucleation site, and $A$ is the nucleation rate[47]. In contrast, considering less porous Super P carbon containing cell in Fig. 2a with a lower nucleation rate, the initial density of nuclei increases linearly with time: $N = N_o A t$ (i.e. progressive nucleation)[48,49]. Moreover, the porous structure of KB also accelerates the dissolution

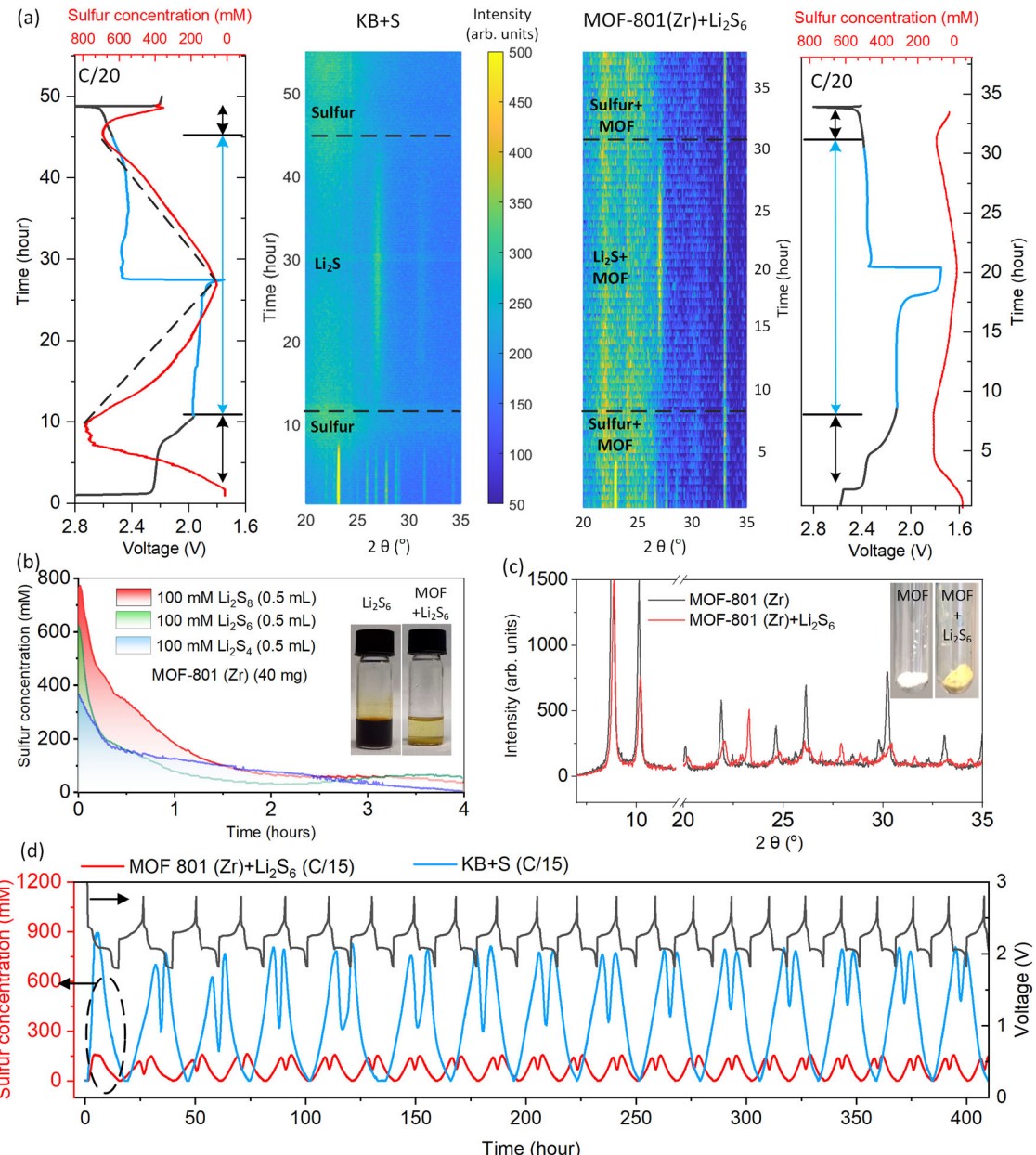

**Fig. 5 | Chemistry of LSB with polysulfide-trapping capability cathode.**
**a** Operando measurement of LSB by TFBG and XRD at C/20 (0.26 mA, left panel: nonpolar physical adsorption of polysulfide by KB carbon; right panel: polar adsorption of polysulfide by MOF-801(Zr)). **b** In-situ detection of polysulfide adsorption by MOF-801(Zr). **c** XRD pattern for MOF-801(Zr) before and after adsorption of $Li_2S_6$. **d** The temporal voltage (C/15, 0.375 mA, gray line), decoded sulfur concentration (red line) of cathode composite based on MOF-801(Zr), and decoded sulfur concentration (blue line) considering the cathode with KB substrate (same amount of active material, used as a reference without showing the temporal voltage).

and re-crystallization of sulfur, which results in a strong sulfur concentration saturation at the transient stage relating to semisolid $Li_2S_4$ generation (keeping a balance between electrochemical and disproportionation process) and enhanced soluble sulfur reduction to solid sulfur at the end of charging (Supplementary Fig. S11). The dynamic response of sulfur concentration to potential and cycling rate is consistent with the Super P substrate, detailed in Supplementary Fig. S12.

To further extend the use of porous materials as host matrix for sulfur confinement, Metal-Organic Frameworks (MOFs) can be considered as efficient candidates for the selective adsorption of polysulfide species[45,50]. We have considered MOF-801(Zr), a microporous zirconium fumarate with pores of about 5–12 Å and a high specific surface area (1020 (±20) m²/g), to be considered as an efficient

candidate for the adsorption of polysulfide species. Its 3D cubic structure is built-up from $Zr_6O_4(OH)_4$ oxo-clusters linked to fumarate ligands and exhibits abundant missing linkers defect sites and polar reactive terminal –OH groups. Due to the Lewis acidic character of the Zr nodes and the high reactivity of these terminal groups, the polysulfide species (soft Lewis bases) are expected to interact strongly with the framework[51,52]. Depicted in Fig. 5b, the 0.5 ml 100 mM polysulfide $Li_2S_x$ (x = 4, 6, 8) was monitored in real time by fiber sensor during adsorption, which is evidently finished within 1 h ensuring efficient adsorption inside the cell. After fully adsorbing 100 mM $Li_2S_6$ (Fig. 5c), the XRD pattern shows the appearance of new peaks in addition to the ones of MOF-801(Zr), matching with the peaks of pure solid sulfur (Supplementary Fig. S13) consistent with the yellow color of the powder. This explains why the operando test starts with the peaks of

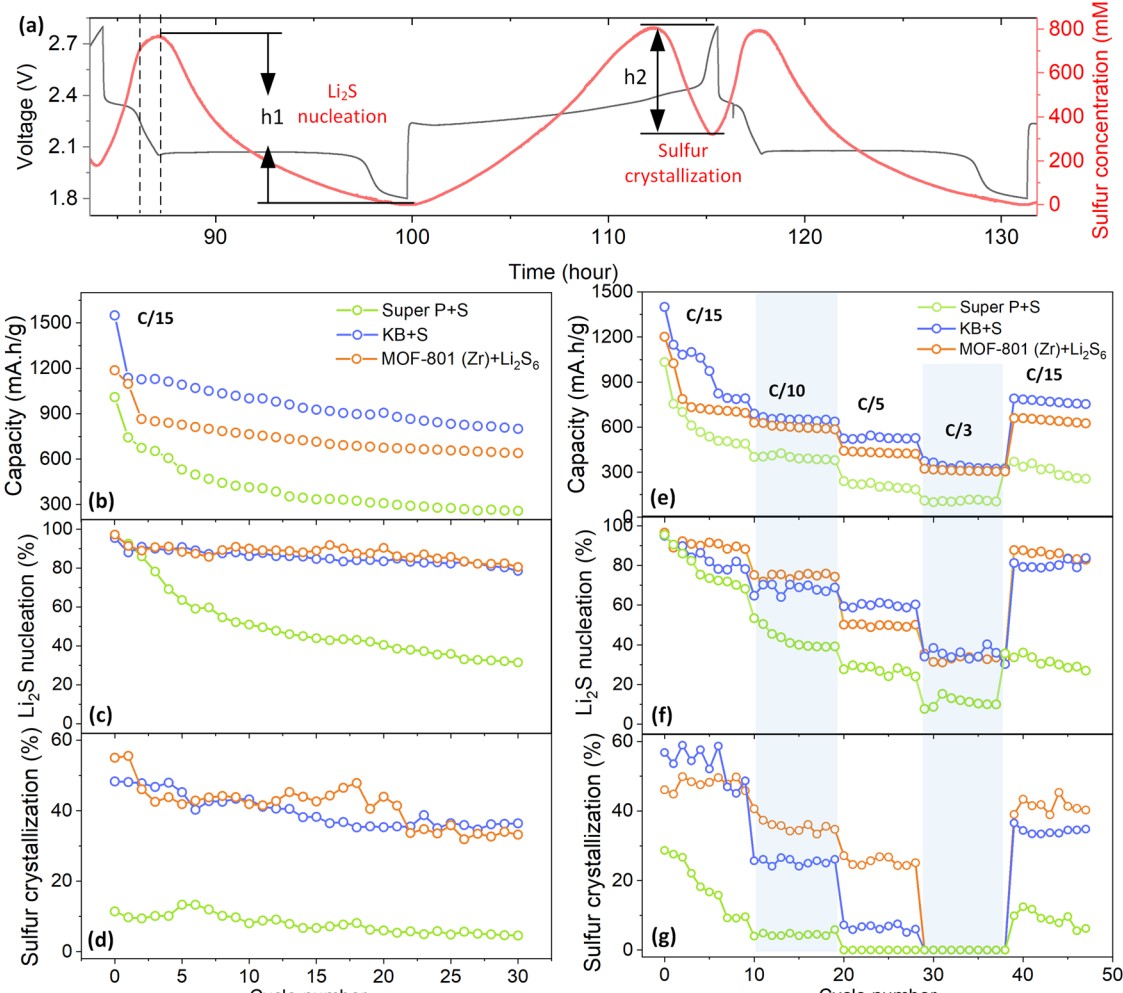

**Fig. 6 | The correlation between electrochemical performance and Li₂S nucleation, sulfur crystallization. a** Content of Li₂S nucleation (h1) and solid sulfur crystallization (h2). **b** Cycling performance of cathode composite at C/15 (0.371 mA) over 30 cycles. **c**, **d** The corresponding ratio of Li₂S nucleation (**c**) and solid sulfur crystallization (**d**). **e** Cycling rate performance of cathode composite (C/ 15, 0.368 mA; C/10, 0.553 mA; C/5, 1.106 mA; C/3, 1.502 mA). **f**, **g** The corresponding ratio of Li₂S nucleation (**f**) and solid sulfur crystallization (**g**). All the cycling cell were pursued in the presence of TFBG fiber, and the Li₂S (h1) and solid sulfur (h2) is normalized by comparing the consumption sulfur in current cycle to that of sulfur fully dissolved in first cycle.

sulfur (XRD pattern on the right panel of Fig. 5a). The subsequent electrochemical reaction is the same as the cathode substrate with KB including concentration saturation in the transition region between the first and the second plateau, nonlinear sulfur consumption rate with instantaneous nucleation pathway and enhanced sulfur crystallization at the end of charge. However, a key point and difference is that the sulfur concentration inside the electrolyte dramatically decreases by 80.8 % with the same amount of active material due to the enhanced adsorption and localization of polysulfides through abundant polar sites in Lewis acid-base chemical interaction by MOF-801(Zr) (Fig. 5d and Supplementary Fig. S14).

To evaluate the performance of cathode considering the three composites (with Super P, KB, MOF-801(Zr)), we decided to compare their long cycling performance together with capabilities of Li₂S nucleation (h1) and sulfur crystallization (h2) (Fig. 6a). Depicted in Fig. 6b, the KB and MOF-801(Zr) cell both experienced better capacity retention over 30 cycles at C/15 with a fade rate as 0.99 %, 0.88% per cycles, respectively, while the super P cell is fading with 4.7% per cycles, which is consistent with the capability of crystallization of Li₂S and sulfur shown in Fig. 6c, d. Surprisingly, the KB and MOF-801(Zr) cells have the similar efficiency of solid species crystallization (also supported by Fig. 6f) by taking the merit of high surface area, whereas

the MOF-801(Zr) exhibits the best capacity retention due to its strong chemical interaction for polysulfide which also well explains the capability to trap and recrystallize sulfur (Fig. 6g). Meanwhile, less sulfur utilization will be induced as well by intrinsic adsorption together with its low conductivity nature, leading to the lower capacity of MOF-801(Zr) than that of KB cell over cycling. On the other hand, the Super P cell is fading fast because of high electrolyte volume with the E/S ratio over 100 μL/mg, presumably due to shuttle phenomenon enhancement. However, this problem does not affect the performance of KB and MOF-801(Zr) cells, revealing that perhaps the most important parameter for cycling performance is related to the capability of crystallization of Li₂S and sulfur, which relies on a comprehensive balance with all involved factors such as high surface area, sulfur anion bonding and conductivity.

## Discussion

Herein, TFBG sensors with low cost, easy integration into batteries, and long cycling capabilities during cell operation were demonstrated for operando testing of electrolyte sulfur concentration evolution of the LSB, which reveals the correlated relationship between the capacity fading and dynamic of dissolution/precipitation of polysulfides over cycling and at different cycling rates. Meanwhile, the chemical kinetics

and thermodynamic response of soluble polysulfide in electrolyte were decoded with GITT experiments, allowing for the dynamic disproportionation process to be linked to the net transport flux of soluble species. Moreover, the cycling performance is well improved by designing the sulfur composite electrode via porous carbon as nonpolar physical sulfur confinement and MOF-801(Zr) as polar adsorption of polysulfide, through a host-guest chemical interaction. These substrates indicate that the nucleation kinetics and growth of $Li_2S$ are changing from a progressive to an instantaneous pathway due to enhanced soluble sulfur consumption rate, ultimately leading to the improvement of crystallization capability of $Li_2S$ and sulfur.

Despite the encouraging insights supported by TFBG technology, a limitation of our operando testing stems from the refractive index which is an average effect of all species inside electrolyte. Thus, somewhat complicated data inference is required with critical external inputs, as comparing to direct evidence and measurement of species by infrared fiber operando measurement[53] and Raman spectroscopy based on hollow-core optical fiber[54]. This could might be solved in future by machine learning algorithms to simplify analysis and hence lead to more efficient data treatment. Second, the optical interrogator used here is both expensive and bulky (volume), so future efforts may warrant an optimized integrator system capable of capturing a sensing signal by energy (amplitude) instead of wavelength[55], but intensity-based measurements may present other technical challenges. Finally, the cycling based on Swagelok is achieved with high E/S ratio which enhances the shuttle effect of polysulfide, leading to acceleration of capacity fading. Therefore, long-term cycling configurations such as coin or 18650 cells integrated with fiber sensors might reveal other prominent performance-governing mechanisms. Nevertheless, TFBG sensors still provide fruitful details of the chemical dynamics of polysulfide as a diagnostic technique to monitor the state of health of cells in real-time.

Considering the specific properties of TFBG (multi-resonance-peaks, high refractive index sensitivity, and ultrafast response), new opportunities of battery sensing can be envisioned such that more than two gratings can be integrated inside the cell to map the sulfur concentration gradient, ultrathin solid electrolyte interphase (SEI) films could be characterized by sensitivity enhanced surface plasmon resonance based TFBG via surface and bulk refractive index discrimination[56], the dynamic of electrons and phonon coupling inside cathode could be probed by ultrafast measurement through the pump-probe configuration of TFBG[57]. Overall, the non-disruptive diagnostic techniques based on the TFBG sensor allow us to monitor chemical-physical-thermal metrics *in operando* with notable time and spatial resolution. Therefore, with the use of these combs it becomes possible to detangle hidden high-value information such as states of charge, health estimations, and operational guidance along with non-electrochemical early-failure indicators, leading to straighter pathways to improving battery reliability, service life, and safety.

## Methods
### Materials and electrode preparation
*Synthesis of MOF-801(Zr)*. 10.86 mmol of $ZrOCl_2 \cdot 8H_2O$, 7.624 mmol of fumaric acid, 9 mL of formic acid and 40 mL of deionized water were mixed in the reactor[58], following 5 h stirring when the solution becomes cloudy. The ultimate product was collected by centrifugation, abundantly washed with water and ethanol, and dried under vacuum.

**Preparation of polysulfides solutions**. The 100 mM lithium polysulfides solution $Li_2S_x$, ($x = 2, 3, …, 8$) were prepared by mixing lithium sulfide (99.9 % $Li_2S$, Sigma Aldrich) and sulfur (S, Sigma Aldrich) in stoichiometric ratio to organic electrolyte (1 M LiTFSI, 0.5 M $LiNO_3$ in DOL/DME (1:1, v/v)), and the solution was continuously stirred with additional heating process at 55 °C for 4 days in argon filled glovebox.

**Preparation of sulfur composite electrodes**. Sulfur and Super P conductive carbon (Ketjen-black carbon) with a ratio 60/40 wt% were mixed by hand-grinding, followed by a heat-treatment (160 °C during 8 h under air). The 5 % wt Poly(tetrafluoroethylene) dispersion (PTFE, Alfa Aesar) is mixed with the cathode composite, rolled to a film and punched into disks (1.2 cm diameter) with sulfur loading around 5.3 mg/cm², and dried under vacuum at 80 °C overnight. To make the cathode composite with MOF-801(Zr), the thoroughly adsorption of 100 mM $Li_2S_6$ in DOL/DME (1:1, v/v) was achieved by MOF-801(Zr) with a stoichiometric ratio of 1 mL/40 mg, followed by washing the powder twice by DOL and tried in vacuum overnight. The sulfur contained MOF-801(Zr) (40% sulfur) was mixed with Super P conductive carbon with 50/50 wt% by hand-grinding without heat-treatment.

### TFBG fabrication and sensing system
Each 10 mm-long, 556.015 nm period TFBG with 7° internal tilt angle was inscribed in hydrogen-loaded CORNING SMF-28 fiber (core diameter: 8.2 µm; clad diameter: 62.5 µm, attenuation: 0.05 dB/km at 1550 nm wavelength) by laser irradiation based on phase-mask method[22]. Hydrogen loading of the fibers, enhancing their photosensitivity to ultraviolet light, was performed at room temperature and a pressure of 15.2 MPa for 14 days. The input light from KrF pulsed excimer laser (model PM-848 from Light Machinery, Inc., emitting at 248 nm and 100 pulse/s) was cylindrically focused along the fiber axis with energy of ~40 mJ over the grating region and also having passed through a 1078.4 nm period phase mask to produce a permanent periodic refractive index modulation in the core of the fiber. Rotating the fiber and phase mask, the tilt of grating fringes was obtained at an angle in the core as 7°.

### Computational details
The transmission spectra simulations were carried out based on three-layer cylindrical waveguide using analytical method[57]. First, the simulated spectra is calibrated by the experimental spectra in air; Second, the refractive index of electrolyte were obtained by increasing the simulation surrounding RI (third layer) manually to match the experimental spectra, which is 1.3858 at 1559 nm wavelength considering the dispersion (supplementary Fig. S2d, e); Finally, mode intensity profiles were simulated by a complex finite-difference vectoral simulation tool (FIMMWAVE, by Photon Design), consisting of a three-layer waveguide: 8.2 µm core diameter with refractive index 1.449311, 125 µm cladding diameter with refractive index 1.444078, 80 µm diameter medium of electrolyte (refractive index = 1.3858).

### Integration of TFBG sensors into modified Swagelok
A ring made of PEEK (12.8 mm diameter, 2 mm thick to fit 10 mm length fiber sensor, storing 250 µL of electrolyte for immersion and electrochemical testing) is fixed in the middle of 19 mm diameter Swagelok cell where fiber sensor can go through by drilling two holes. The Li metal foil (0.38 mm thickness, 14 mm diameter) is attached to one side of PEEK ring as anode, and on the other side of ring there is a steel grid to hold one Whatman separator beneath cathode composite. The cells were assembled in an argon-filled glovebox.

### Electrochemical measurements
The electrochemical performances of Swagelok cell were evaluated by BCS-810 or MPG2 potentiostat (Biologic, France) at 25 °C degree inside temperature-controlled oven (Memmert, ±0.1 °C). The galvanostatic discharge–charge cycling was carried out with the voltage range of 1.7 V-2.8 V.

### Operando measurement by TFBG sensors and XRD
To achieve TFBG sensor operando measurement, the optical transmission spectra were recorded (1 min/spectra) by an optical integrator (CTP10, EXFO SOLUTIONS) with a resolution of 1 pm for wavelength

ranging from 1500 nm–1600 nm. Considering XRD operando measurement, it was performed on a D8 Advance diffractometer (Bruker) using a Cu Kα X-ray source ($\lambda_{K\alpha1}$ = 1.54056 Å, $\lambda_{K\alpha2}$ = 1.54439 Å) and a LynxEye XE detector. The XRD pattern and electrochemical data are simultaneously recorded by a custom designed airtight cell with a beryllium window producing a full pattern every 20 min.

## Preparation of cycled electrode samples for SEM and EDX imaging

The samples were prepared by washing the cathode powder twice by DOL to remove any dissolved lithium polysulfide species and lithium salt, and dried in vacuum chamber overnight. Then the powder was coated with gold (plasma sputtering coater (GSL-1100X-SPC-12, MTI)) for SEM (FEI Magellan) equipped with an energy-dispersive X-ray spectroscopy detector (Oxford Instruments) performed under an acceleration voltage of 20 kV.

## Data availability

The origin data generated in this study are provided in the Source data file. Extra data are available on request from the corresponding author. Source data are provided with this paper.

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

## Acknowledgements

J.-M.T. acknowledges the International Balzan Prize Foundation and the LABEX STOREX II for funding. F.Liu and J.-M.T. acknowledge the European Project "Innovative physical/virtual sensor platform for battery cell" (INSTABAT) (European Union's Horizon 2020 research and innovation program under grant agreement No 955930). W. Lu acknowledges the support of the CSC scholarship (201906880002). R.D.-C. is thankful to the French Embassy for the Visiting Researcher Fellowship (135694 V). We thank Dr. J. Forero-Saboya for his assistance of scanning electron microscopy images. We thank Dr. F. Betermier for her assistance of preparing cathode. We thank Prof. J. Albert from Carleton University (Ottawa, Canada) for the fabrication of fiber sensors and assistance of simulation software. Finally, we gladly thank Dr. W. He, Dr. Y. Wang, Dr. X. Gao, Dr. B. Li, Dr. C. Gervillié-Mouravieff and Mr. C. Leau for extensive and valuable discussion and comments.

## Author contributions

F.L., J.H., R.D.-C. and J.-M.T. conceived the idea and designed the experiments. F.L. performed the experiments. F.L., J.H., R.D.-C. and J.-M.T. performed the data analysis. W.L. and V.P. provided the MOF-801(Zr). Finally, F.L., S.B., R.D.-C. and J.-M.T. wrote the paper with contributions from all authors.

## Competing interests

The authors declare no competing interests.
