## [Peer Review File · Nature Communications]

REVIEWER COMMENTS

Reviewer #1 (Remarks to the Author):

In this work, TFBGs were employed to operando track the chemical dynamics/states of the Li-S battery via electrolyte sulfur concentration, revealing the correlated relationship between the capacity fading and dynamic of dissolution/precipitation of polysulfides over cycling and at different cycling rates. Overall, I think is an interesting work, since the authors proposed a new application of optical fiber sensor for battery monitoring. Consequently, I will recommend its publication after a minor revision.

(1) In Fig1c, why did the author choose the “*” region as the sensing mode? Can other modes also be used as sensing modes, and if so, what are the differences in sensing performance?

(2) As we all know, the S cathodes exhibit around 80% volume changes during cycling. Therefore, in this manuscript, please explain whether the wavelength shift caused by the volume changes of the S cathode will affect the results.

(3) In manuscript, the author filled 500 μL of electrolyte into the cell, which is different from the common application scenario of optical fiber (the fiber is placed in a huge amount of liquid). Therefore, the author should consider whether the electrolyte was sufficient to completely infiltrate the fiber and can evenly wrap the optical fiber, which will affect the result.

(4) Ultimately, optical fiber is a linear sensor with a limited detection range. In manuscript, the author also mentioned that the “delay” due to position of the sensor in the cell. So can the author show the evolution of sulfur concentration in other locations, such as near the electrode or at the same level as the current fiber?

(5) In Fig. 4b, I observed a slight dissonance in 12th cycle where the drop in temperature caused the wavelength to decrease. However, the double effect of temperature dropping and Li_2S nucleation will result in smaller wavelength. Why is the valley of sulfur concentration in 12th cycle greater than that in other cycles?

(6) In Fig. 4b, an increase of the background of sulfur concentration was attributed to the strong shuttle effect with less sulfur utilization caused by the high E/S ratio. However, other components in the electrolyte also undergo irreversible chemical reactions during cycling (> 400 h). So how can you confirm that the increase of the background of sulfur concentration is solely due to the strong shuttle effect with less sulfur utilization?

(7) KB was better than SP as physical nonpolar sulfur confinement host. Therefore, the sulfur concentration in KB/S should be smaller than SP/S. In manuscript, the sulfur concentration in KB/S was ~ 700 mM and the sulfur concentration in SP/S was ~ 500 mM. Please explain it.

(8) Typos or hard to understand. The authors need to rephrase the following parts.

-In abstract, "...the nucleation pathway and crystallization of Li₂S and sulfur governs the cycling performance..." should be "...the nucleation pathway and crystallization of Li₂S and sulfur govern the cycling performance..."

-On page 1, "...and shuttle effect caused by soluble polysulfide in electrolyte..." should be "...and the shuttle effect caused by soluble polysulfide in electrolyte..."

-On page 11, "...On the upper voltage plateau the solid sulfur dissolution or recrystallization..." should be "...On the upper voltage plateau, the solid sulfur dissolution or recrystallization..."

-On page 15, "...changing from progressive to an instantaneous pathway ..." should be "...changing from a progressive to an instantaneous pathway ..."

Reviewer #2 (Remarks to the Author):

The paper entitled "Detangling electrolyte chemical dynamics and evolution in Li-S batteries by operando monitoring with optical resonance combs" reports about TFBG technique for studying electrolyte chemical dynamics and evolution in Li-S batteries by tracking sulfur concentration and demonstrate that the nucleation pathway and crystallization of Li₂S and sulfur governs the cycling performance. Although the similar technique has already been used in their previous work to track electrolyte concentration through refractive index (DOI: 10.1039/d1ee02186a), this work brings a new insight on understanding the mechanism and electrolyte chemical dynamics of LSB. Thus, this paper could be considered for publication after a minor revision, detailed in the following comments:

1. Why the operando XRD in fig 2a doesn't show the recrystallization process of sulfur at the end of charging?

2. There is a sudden drop of sulfur concentration from A to B in the beginning of discharge in fig 3c, which also happens at the beginning of next discharge period. What leads to this? The sulfur concentration from A to B in fig 3c reacts different from in fig 3a and b. It doesn't seem to be attributed to "delay".

3. The soluble sulfur transport flux calculation should be based on equilibrium state, as the result is used for representing the whole process. So I think it is more reasonable to exclude region from A to B and C to D during transport flux calculation. Then the net transport flux should be $VBC/(S \times t)$.

4. The explanation of green triangle in fig 3e is "Transport flux of soluble sulfur based on current pulse (kinetics process)". Actually, the transport flux of soluble sulfur based on current pulse is $(VBC - VDE) / (S \times t)$, because the electrochemical process is accompanied by disproportionation process. Or the green triangle should be described as the net transport flux of soluble sulfur.

5. Adding a fig comparing cycling performance under three cycle mode in fig 4e would clearly show how the five stage affect battery performance and enhance persuasion of this part.
6. Please explain how you control internal temperature as well as strain stable during charging and discharging processes to eliminate their influence on the wavelength shift.
7. Could you please specify the definition of the ratio which indicates the consumption rate of sulfur under the galvanostatic condition on Page 7? Furthermore, the sulfur concentration variation rate corresponding to each plateau would better be given in Fig.S4a.
8. In “TFBG fabrication and sensing system”, please add the specific structure of the TFBG sensor employed in the measurements should be introduced, including the core/cladding diameter, the coating material, the period of grating and so on.
9. In supplementary video, the amplitude and the wavelength have same trends. What is the difference between them? Can we draw the same conclusion from amplitude instead of wavelength?
10. How to achieve optical fiber inserted into the battery without leaking electrolyte and how to make sure there is no side effect or carryover effect due to inserted fibre?
11. How reproducible is the experiment ?

Reviewer #1:

In this work, TFBGs were employed to operando track the chemical dynamics/states of the Li-S battery via electrolyte sulfur concentration, revealing the correlated relationship between the capacity fading and dynamic of dissolution/precipitation of polysulfides over cycling and at different cycling rates. Overall, I think is an interesting work, since the authors proposed a new application of optical fiber sensor for battery monitoring. Consequently, I will recommend its publication after a minor revision.

Author response: We highly appreciate the positive comments from the reviewer, and they are all considered in corrected manuscript.

Question 1: In Fig1c, why did the author choose the “” region as the sensing mode? Can other modes also be used as sensing modes, and if so, what are the differences in sensing performance?*

Author response: The optical fiber sensor resonance marked by “*” was chosen as the preferred sensing mode due to the response sensitivity and particular optical polarization property.

Regarding the sensing sensitivity, the primary advantage of the chosen resonance is that, for the accessible spectrum, it shows largest refractive index sensitivity (and hence response to sulfur concentration) compared to other guided modes at longer wavelengths that could also be considered [R1]. It is also referred to as the “cut-off” mode, where the surrounding refractive index (i.e. concentration of polysulfide dissolved in electrolyte) becomes equal to its mode effective index, and thereby it is observed that the mode resonance shifts more rapidly as its evanescent field penetrates more into the outer medium (electrolyte). With the goal of collecting valuable details regarding polysulfide chemical dynamics and evolution in electrolyte, the cut-off mode marked by “*” should be the best choice.

Turning to the optical polarization of the relevant guided mode, it is well known that there are two group resonances in transmission spectra: P-polarization (blue line in Fig. 1(a)) corresponding to guide mode electric field azimuthally polarized; S-polarization (red line in Fig. 1(a)) corresponding to guided mode electric field radially polarized [R2]. Given the fact that the cutoff mode (marked by “*”) will be polarization insensitive (See Fig 1b, 1c insets) and there is no wavelength shift except some amplitude variation, it inspires confidence that it is possible to decouple the sulfur concentration in electrolyte by tracking the wavelength shift of cutoff mode with un-polarized input light and without a polarizer. By doing so, it allows for a simplified sensing system which still ensures that detection is both stable and repeatable.

Fig. 1 | Optical polarization property of TFBG: (a) Experimental polarized transmission spectra in electrolyte (red, S-pol input, and blue, P-pol input); (b) Radial (P-pol) and (c) azimuthal (S-pol) dependence of simulated E-field intensity of cut-off guided mode (the arrows show the E-field vector orientations in both cases).

[R1] Chan, C.-F., Chen, C., Jafari, A., Laronche, A., Thomson, D. J. & Albert, J. Optical fiber refractometer using narrowband cladding-mode resonance shifts. *Appl. Opt.* 46, 1142-1149 (2007).

[R2] Alam, M.-Z., Albert, J. Selective excitation of radially and azimuthally polarized optical fiber cladding modes. *J. Lightw. Technol.* 31, 3167-3175 (2013).

Question 2: As we all know, the S cathodes exhibit around 80% volume changes during cycling. Therefore, in this manuscript, please explain whether the wavelength shift caused by the volume changes of the S cathode will affect the results.

Author response: The reviewer has rightfully noted a particular challenge with the Li-S system, but this is something where our sensor design has a particular advantage. The volume changes of sulfur during cycling (both expansive and contractive) don't affect the wavelength shift of our sensor due to the specific cell design, mentioned in Fig. S3 in supplementary information. Basically, we are first placing a 2 mm thick, 12.8 mm diameter polyether ether ketone (PEEK) spacer ring into the swagelok assembly. This ring is pierced in the middle such that we can inject the fiber (which has a 1 cm long TFBG sensor inscribed segment) into the cross-sectional center of the cell. Within the Swagelok, the PEEK ring separates the cathode (sulfur and Super P carbon composite (60/40 wt.%) and lithium anode so that fiber sensor is perfectly immersed inside the electrolyte but not touching, nor at risk of touching, either electrode regardless of the respective volume change.

During our experiments, we ensure temperature and strain remain effectively constant because 1) the slow cycling rate gives minimal heat generation from overpotential and did not result in any temperature fluctuation of the sensor; additionally noted is that the cells are placed in a well-regulated thermostatic oven and 2) with the TFBG being effectively isolated and solely in the liquid electrolyte, it should not be sensitive to strain related to the electrodes. This is the particular case here, as we are using a high sulfur ratio (E/S ratio, $\sim 100 \mu\text{L}/\text{mg}$).

Question 3: In manuscript, the author filled 500 μL of electrolyte into the cell, which is different from the common application scenario of optical fiber (the fiber is placed in a huge amount of liquid). Therefore, the author should consider whether the electrolyte was sufficient to completely infiltrate the fiber and can evenly wrap the optical fiber, which will affect the result.

Author response: We fully agree with the reviewer's consideration that the fiber must be fully immersed in the electrolyte, which is the key point for ensuring that the sensor works properly. As mentioned above, a PEEK ring is used in the Swagelok assembly stack, and it is through the middle of this ring where the TFBG sensor is inserted. The volume inside this PEEK ring serves as a container, which is filled electrolyte. As the ring thickness of 2 mm is much greater than the fiber diameter of 0.125 mm, the PEEK ring provides a nice pool of liquid to constantly immerse the fiber during cell operation. To help illustrate the geometric considerations, Fig. 2 is shared below.

Fig. 2 | The PEEK container.

Question 4: Ultimately, optical fiber is a linear sensor with a limited detection range. In manuscript, the author also mentioned that the “delay” due to position of the sensor in the cell. So can the author show the evolution of sulfur concentration in other locations, such as near the electrode or at the same level as the current fiber?

Author response: The question regarding the position of sensors is well spotted. This is an important aspect that we have also explored because any significant ion transport latency would have important implications for understanding dynamic reactions, as is the case here.

Regarding our terminology, “delay” is used to refer to the time needed by the new polysulfide species generated at the positive electrode to lead an equilibrium state within the electrolyte. To quantify the extent of “delay”, a further experiment was carried out by placing two TFBGs sensors at different positions within the volume of electrolyte. This allows us to see the sulfur concentration dynamics close to Li metal and sulfur electrode surfaces, respectively. As shown in Fig. 3, the sulfur concentration detected by TFBG1, closer to the surface of Li (red line), typically “falls behind” that of the concentration detected by TFBG2, near to the surface of sulfur (blue line). A second and notable change is the lower amplitude of the S concentration near the surface of Li, as compared to the blue curve associated with the polysulfide of the positive electrode. While certainly the reviewer is aware of the value hidden within the details of the two sensor signals, given their positions, further assessment is beyond the scope of the work reported herein. Nevertheless, the position dependent differences observed here are small and do not impact our overall conclusions nor understanding of mechanisms.

Fig. 3 | Decoding sulfur concentration gradient of LSB by two sensors close to electrode.

Question 5: In Fig. 4b, I observed a slight dissonance in 12th cycle where the drop in temperature caused the wavelength to decrease. However, the double effect of temperature dropping and Li₂S nucleation will result in smaller wavelength. Why is the valley of sulfur concentration in 12th cycle greater than that in other cycles?

Author response: We thank the reviewer for this keen observation and would firstly like to apologize for the confusion stemming from this anomaly. The “valley” mentioned by the referee is not a “real” wavelength shift induced by a temperature change or generated polysulfides. It is the result of an error in plotting the data which we should have spotted ourselves. This anomaly comes from a recording failure of the integrator software that lasted for about 10 hours due to Windows update of computer system, and during this time period there was no data recorded. We have replotted Fig. 4b in the main text with “dot” instead of “line”, which we hope makes sense in terms of the wavelength shift.

Regarding the second remark by the referee about the double effect of temperature, thermal effects can be totally removed by a thorough thermal calibration process that enlists several steps as follows:

Step 1: By testing the temperature response of the fiber sensor in air (Fig. 4a,b), the thermal sensitivity of the core mode is determined to be 10.2 pm/°C and the target cladding mode sensitivity (cutoff mode in electrolyte) is 9.7 pm/°C (the cladding mode thermal sensitivity is always smaller than that of core mode) [R3].

Step 2: By testing temperature response of the fiber sensor immersed in electrolyte (Fig. 4c,d), the total wavelength shift of the cutoff mode comprises two parts: temperature (9.7 pm/°C obtained from step 1) and the temperature-modulated refractive index of the electrolyte (-9.5 pm/°C in Fig. 4d).

When the cell is cycled with the fiber sensor, the observed wavelength shift will be composed of temperature, temperature-induced refractive index, and polysulfide-induced refractive index of electrolyte. By manually compensating for the thermal effects on the basis of steps 1 and 2, the wavelength shift will be linked solely to the polysulfide generated.

For the consideration of the reviewer and future readers, the above discussions have been added in the revised supplementary information as Fig. 4S.

Fig. 4 | Thermal calibration of electrolyte: (a) The thermal response in air and (b) The corresponding thermal sensitivity of core mode and target cladding mode (cutoff mode in electrolyte); (c) The thermal response in electrolyte and (d) sensitivity.

[R3] Imas, J. J., Bai, X., Zamarreño, C. R., Matías, I. R. & Albert, J. Accurate compensation and prediction of the temperature cross-sensitivity of tilted FBG cladding mode resonance. *Appl. Opt.* 62, E8-E15 (2023).

Question 6: In Fig. 4b, an increase of the background of sulfur concentration was attributed to the strong shuttle effect with less sulfur utilization caused by the high E/S ratio. However, other components in the electrolyte also undergo irreversible chemical reactions during cycling (> 400 h). So how can you confirm that the increase of the background of sulfur concentration is solely due to the strong shuttle effect with less sulfur utilization?

Author response: The reviewer has made another interesting suggestion concerning the increase in background noise, in addition to the lower sulfur usage caused by the high E/S ratio. It is certainly true that other irreversible reactions processes such as the formation of a solid electrolyte interphase (SEI) by the consumption of the additive LiNO_3 , and the decomposition of the solvents DOL and DME [R4] could affect the sensing results by altering the interlinking refractive index of the electrolyte. However, this is exactly the reason why we have pursued a high E/S ratio so that the electrolyte consumption due to SEI formation is quite limited by using small amount of active material comparing to electrolyte (5-6 mg sulfur and 500 μL electrolyte). To us, this is well-confirmed by the experiment (Fig. 5d in the main text) involving the cathode composite based on Ketjen black carbon (KB) or MOF-801(Zr), which shows that the sulfur concentration background is fairly stable during cycling due to the higher efficiency of the nucleation pathway and crystallization of Li_2S .

[R4] Xiong, S., Xie, K., Diao, Y & Hong, X. Characterization of solid electrolyte interphase on lithium anode for preventing the shuttle mechanism in lithium-sulfur batteries. *J. Power Sources* 246, 840-845 (2014).

Question 7: *KB was better than SP as physical nonpolar sulfur confinement host. Therefore, the sulfur concentration in KB/S should be smaller than SP/S. In manuscript, the sulfur concentration in KB/S was ~700 mM and the sulfur concentration in SP/S was ~500 mM. Please explain it.*

Author response: We totally agree with the reviewer's argument that theoretically higher surface area carbon (KB) should lead to trapping more sulfur. There could be several explanations for these discrepancies:

To start with, both carbons are non-polar and the adsorption ability is very small compared to, for example, oxygenated porous architectures [R5]. Inspired by the reviewer's comment, we tested the adsorption ability of SP and KB carbons by mimicking the similar sulfur/carbon ratio we have used in our cells. Both carbons showed almost no adsorption ability, which was visually detected after resting for 22 hours (Fig. 5).

Fig 5. The adsorption test of polysulfide by SP or KB carbon.

In addition to that, as was pointed in Fig. S10 in supplementary information with the XRD patterns before and after heat treatment, sulfur only partially penetrated into the nanostructure of KB, contrary to our own expectation, as well as literature reports. Moreover, physical nonpolar sulfur confinement by Ketjen black (KB) is very limited in our experiment, supported by Fig. S11 in supplementary information there is no sulfur left tested by energy-dispersive X-ray spectroscopy (EDX) inside the KB after the first plateau of discharge. We must therefore infer that all sulfur is converted to polysulfide and dissolved in electrolyte where it is detected by the fiber sensor.

Another reason may rise from our experimental condition in which we deliberately used a much higher E/S ratio (~100) than usual (< 8) so that the solubility of polysulfides would be feasible even for the highly porous hosts. To help with detection, a relatively small current density (C/20) gives time for dissolved polysulfide to equilibrate in the electrolyte, regardless of how much sulfur is trapped into the pores.

Finally, the slightly higher sulfur concentration of KB/S than SP/S could be due to the fact that the weight of the active material is slightly larger than that of the Super P/S composite disk, which

can be attributed to the manual process of mixing the electrode composite with PTFE, forming it into a film and punching it into disks. As a result, there will be some deviation in the weight of the active material, but this is generally within a controllable range, and doesn't materially impact any of the results presented.

[R5]. Demir-Cakan, R., Morcrette, M., Nouar, F., Davoisne, C., Devic, T., Gonbeau, D., Dominko, R., Serre, C., Ferey, G & Tarascon, J.-M. Cathode composites for Li-S batteries via the use of oxygenated porous architectures. *J. Am. Chem. Soc.* 133, 40, 16154–16160 (2011).

Question 8: Typos or hard to understand. The authors need to rephrase the following parts.

-In abstract, "...the nucleation pathway and crystallization of Li₂S and sulfur governs the cycling performance..." should be "...the nucleation pathway and crystallization of Li₂S and sulfur govern the cycling performance..."

-On page 1, "...and shuttle effect caused by soluble polysulfide in electrolyte..." should be "...and the shuttle effect caused by soluble polysulfide in electrolyte..."

-On page 11, "...On the upper voltage plateau the solid sulfur dissolution or recrystallization..." should be "...On the upper voltage plateau, the solid sulfur dissolution or recrystallization..."

-On page 15, "...changing from progressive to an instantaneous pathway ..." should be "...changing from a progressive to an instantaneous pathway ..."

Author response: We really appreciate the time spent by the referee in identifying and editing inconsistencies found in our submitted manuscript. We have accepted all suggestions and reworded the passages in the text accordingly.

Reviewer #2:

The paper entitled “Detangling electrolyte chemical dynamics and evolution in Li-S batteries by operando monitoring with optical resonance combs” reports about TFBG technique for studying electrolyte chemical dynamics and evolution in Li-S batteries by tracking sulfur concentration and demonstrate that the nucleation pathway and crystallization of Li_2S and sulfur governs the cycling performance. Although the similar technique has already been used in their previous work to track electrolyte concentration through refractive index (DOI: 10.1039/d1ee02186a), this work brings a new insight on understanding the mechanism and electrolyte chemical dynamics of LSB. Thus, this paper could be considered for publication after a minor revision, detailed in the following comments:

Author response: We highly appreciate the positive comments from the reviewer, and they are all considered in corrected manuscript.

Question 1: *Why the operando XRD in fig 2a doesn't show the recrystallization process of sulfur at the end of charging?*

Author response: The reviewer has indeed made a careful observation. We believe it is a question of quantity and state of sulfur (amorphous vs crystallize). From our TFBG decoupling experiment we can deduce that only ~10% solid sulfur will be reformed without providing clues on its state (crystallize or amorphous). Ten percent of crystallized sulfur should be easily detected by XRDs. To check this point that the re-formed sulfur is amorphous, a fully charged sulfur-loaded carbon electrode was recovered by washing and drying to remove any soluble polysulfide as well as remaining electrolyte salts and investigated by SEM. Fig. 1 compares two SEM taken shots of the pristine and fully charge samples suggesting the presence of amorphous sulfur, hence confirming the XRDs. We may also note that sulfur is readily amorphized in the presence of organic compounds under mild conditions, which has been well-established in various sulfur industries going back 70 years, at least [R1]

[R1] Bartlett, P.-D., Meguerian, G. Reactions of elemental sulfur. I. The uncatalyzed reaction of sulfur with triarylphosphines. *J. Am. Chem. Soc.* 78, 15, 3710-3715 (1956).

Fig. 1 | The morphology (SEM) of cathode (S: Super P=6: 4 wt%) at the beginning of discharging and end of charging

Question 2: *There is a sudden drop of sulfur concentration from A to B in the beginning of discharge in fig 3c, which also happens at the beginning of next discharge period. What leads to this? The sulfur concentration from A to B in fig 3c reacts different from in fig 3a and b. It doesn't seem to be attributed to "delay".*

Author response: Regarding the reason leading to the "tiny" sudden drop of sulfur concentration from A to B in Fig.3c during charging (transition from rest mode to cycle mode), this can be attributed to the instantaneous response of electrolyte to a current pulse, arising from the redistribution of polysulfide (concentration gradient) because of sudden electric field. In this way, it is not exactly a "delay", as noted by the referee. While it is quite different for the case in Fig. 3a and b that concentration variation from A to B is opposite to that in Fig. 3c, this should be attributed to the change of the concentration gradient direction during discharge. We also note that the amplitude of concentration variation from A to B is smaller than that in Fig. 3c, resulting from the fact that the fiber sensor is physically/geometrically closer to cathode during assembly process, and therefore it leads to an asymmetric concentration variation.

Question 3: *The soluble sulfur transport flux calculation should be based on equilibrium state, as the result is used for representing the whole process. So I think it is more reasonable to exclude region from A to B and C to D during transport flux calculation. Then the net transport flux should be $VBC/(S \times t)$.*

Author response: We sincerely appreciate that the referee makes the definition of net transport flux clearer ($VBC/(S \times t)$) and we fully agree with the change. It has been corrected in main text of Fig. 3.

Question 4: *The explanation of green triangle in fig 3e is "Transport flux of soluble sulfur based on current pulse (kinetics process)". Actually, the transport flux of soluble sulfur based on current pulse is $(VBC - VDE)/(S \times t)$, because the electrochemical process is accompanied by disproportionation process. Or the green triangle should be described as the net transport flux of soluble sulfur.*

Author response: The referee is perfectly right to assert that the net transport flux of soluble sulfur is based on electrochemical process accompanied by disproportionation process in Fig. 3e. It has been corrected in main text.

Question 5: *Adding a fig comparing cycling performance under three cycle mode in fig 4e would clearly show how the five stages affect battery performance and enhance persuasion of this part.*

Author response: The referee's suggestion is a very good one and accordingly a figure has been prepared regarding cycling performance. It can now be found added to Fig. 4 in main text.

Question 6: *Please explain how you control internal temperature as well as strain stable during charging and discharging processes to eliminate their influence on the wavelength shift.*

Author response: Interestingly, the same question was asked by referee 1 (questions 3 and 5) and we have provided effectively the same answers. To eliminate any deformation induced by the large volume changes (around 80%) of the active material, the cell is specifically designed as mentioned in Fig. S3 in supplementary information, as well as Fig 2 shared again below. Basically, we are first placing a 2 mm thick, 12.8 mm diameter polyether ether ketone (PEEK)

spacer ring into the swagelok assembly. This ring is pierced in the middle such that we can inject the fiber (which has a 1 cm long TFBG sensor inscribed segment) into the cross-sectional center of the cell. Within the Swagelok, the PEEK ring separates the cathode (sulfur and Super P carbon composite (60/40 wt. %)) and lithium anode so that fiber sensor is perfectly immersed inside the electrolyte but not touching, nor at risk of touching, either electrode regardless of the respective volume change.

Fig. 2 | The PEEK container.

During our experiments, we ensure temperature and strain remain effectively constant because 1) the slow cycling rate gives minimal heat generation from overpotential and did not result in any temperature fluctuation of the sensor; additionally noted is that the cells are placed in a well-regulated thermostatic oven and 2) with the TFBG being effectively isolated and solely in the liquid electrolyte, it should not be sensitive to strain related to the electrodes. This is the particular case here, as we are using a high sulfur ratio (E/S ratio, $\sim 100 \mu\text{L}/\text{mg}$).

Any thermal effect during cycling can be totally removed by a thorough thermal calibration process that enlists several steps as follows:

Step 1: By testing the temperature response of the fiber sensor in air (Fig. 3a,b), the thermal sensitivity of the core mode is determined to be $10.2 \text{ pm}/^\circ\text{C}$ and the target cladding mode sensitivity (cutoff mode in electrolyte) is $9.7 \text{ pm}/^\circ\text{C}$ (the cladding mode thermal sensitivity is always smaller than that of core mode) [R2].

Step 2: By testing temperature response of the fiber sensor immersed in electrolyte (Fig. 3c,d), the total wavelength shift of the cutoff mode comprises two parts: temperature ($9.7 \text{ pm}/^\circ\text{C}$ obtained from step 1) and the temperature-modulated refractive index of the electrolyte ($-9.5 \text{ pm}/^\circ\text{C}$ in Fig. 3d).

When the cell is cycled with the fiber sensor, the observed wavelength shift will be composed of temperature, temperature-induced refractive index, and polysulfide-induced refractive index of electrolyte. By manually compensating for the thermal effects on the basis of steps 1 and 2, the wavelength shift will be linked solely to the polysulfide generated.

For the consideration of the reviewer and future readers, the above discussions have been added in the revised supplementary information as Fig. 4S.

Fig. 3 | Thermal calibration of electrolyte: (a) The thermal response in air and (b) the corresponding thermal sensitivity of core mode and target cladding mode (cutoff mode in electrolyte); (c) The thermal response in electrolyte and (d) sensitivity.

[R2] Imas, J. J., Bai, X., Zamarreño, C. R., Matías, I. R. & Albert, J. Accurate compensation and prediction of the temperature cross-sensitivity of tilted FBG cladding mode resonance. *Appl. Opt.* 62, E8-E15 (2023).

Question 7: Could you please specify the definition of the ratio, which indicates the consumption rate of sulfur under the galvanostatic condition on Page 7? Furthermore, the sulfur concentration variation rate corresponding to each plateau would better be given in Fig.S4a.

Author response: We thank the referee for noting that this was not clear in our initial submission. The ratio indicating the consumption rate of sulfur under the galvanostatic condition is defined

by: $Ratio = \left| \frac{\text{first plateau concentration slope}}{\text{second plateau concentration slope}} \right|$. Therefore, the $Ratio_{discharge} =$

$\left| \frac{S1}{S2} \right| = 3.88$, $Ratio_{charge} = \left| \frac{S4}{S3} \right| = 0.84$. A figure regarding to sulfur concentration variation rate

has been added to Fig.S5 in the supplementary information, which should help visually clarify the origin of these ratios.

Fig. 4 | The definition of consumption rate of sulfur under the galvanostatic condition.

Question 8: In “TFBG fabrication and sensing system”, please add the specific structure of the TFBG sensor employed in the measurements should be introduced, including the core/cladding diameter, the coating material, the period of grating and so on.

Author response: We have added more details about the specific structure of TFBG to “TFBG fabrication and sensing system” in main text in accordance with the referee’s request.

Question 9: In supplementary video, the amplitude and the wavelength have same trends. What is the difference between them? Can we draw the same conclusion from amplitude instead of wavelength?

Author response: Indeed, the amplitude and wavelength of TFBG cutoff resonance are strongly correlated by refractive index (sulfur concentration) variation. The cutoff resonance decreases sharply in amplitude together with the wavelength shift (Fig. 5), indicating loss of total internal reflection at the point where the cladding mode effective index becomes equal to the surrounding refractive index of polysulfide solution. Amplitude and wavelength will follow the same trend when the surrounding solution is a “pure” liquid, except in the special case of high turbidity, which leads to the irreversible disappearance of all cladding mode resonances [R3]. Thus, the observation of similar amplitude and wavelength trends in our video means that the effects of turbidity in our system, if any, are negligible.

Fig. 5 | The TFBG response to 100 mM Li_2S_2 , Li_2S_5 and Li_2S_8 in electrolyte of 1 M LiTFSI, 0.5 M LiNO_3 in DOL/DME (1:1, v/v)

[R3] Huang, J., Han, X., Liu, F., Gervillié, C., Blanquer, L. A., Guo, T. & Tarascon, J.-M. Monitoring battery electrolyte chemistry via in-operando tilted fiber Bragg grating sensors. *Energy Environ. Sci.* 14, 6464-6475 (2021).

Question 10: How to achieve optical fiber inserted into the battery without leaking electrolyte and how to make sure there is no side effect or carryover effect due to inserted fibre?

Author response: As shown in Fig. 2 we have used a Swagelok cell design for our experiments, which is nearly a worldwide standard in battery research labs because of its relative ease of assembly and air/moisture tightness. Swagelok ferrules typically rely on plastic deformation to ensure excellent sealing properties. The cell is then diametrically drilled through the PEEK spacer to accommodate the fiber that supports the TFBG and is hermetically sealed with epoxy at the fiber entry and exit positions in the Swagelok assembly. With this method we have never experienced electrolyte leakage.

As far as secondary or carry-over effects due to the inserted fiber are concerned, they are minimal here because the fiber is immersed in an electrolyte bath (PEEK ring) and is largely separated from the positive and negative electrode. Of course, the story would have been very different if we had placed the fiber in the sulfur electrode, due to the limitations of ion transport and induced current inhomogeneity!

Question 11: How reproducible is the experiment?

Author response: These experiments enlist several key steps out of which three, namely i) the preparation of consistent C-S electrodes ii) the proper positioning of the fiber and iii) the feasibility of having similar TFBG sensors, were found to be the most critical for overall reproducibility. Nevertheless, we could well-master the first two steps in-house and the third one by working with our TFBG's producer, such that highly reproducible and dependable data could be obtained. Overall, both sensors and cells needed to be developed to a point that we can ensure our data is repeatable and reproducible (with respect to sulfur concentration evolution during cell operation). As per the question of the referee we should have addressed this point by adding the sentence "all the data has been at least be duplicated 2 or 3 times prior to being reported", which has now been included in the main text.

REVIEWERS' COMMENTS

Reviewer #1 (Remarks to the Author):

The authors have well addressed the previous concerns. I think this paper can be accepted.

Reviewer #2 (Remarks to the Author):

The manuscript can be accepted.